# Action Imitation in Common Action Space for Customized Action Image Synthesis

**Wang Lin**[1,*]**, Jingyuan Chen**[1,†]**, Jiaxin Shi**[2,*]**, Zirun Guo**[1]**, Yichen Zhu**[1]**, Zehan Wang**[1]**,**
**Tao Jin**[1]**, Zhou Zhao**[1]**, Fei Wu**[1]**, Shuicheng YAN**[3]**, Hanwang Zhang**[4]
Zhejiang University[1], Xmax.AI[2], Skywork AI Singapore[3], Nanyang Technological University[4]
{linwanglw, jingyuanchen, gzr, yc_zhu, wangzehan01,
jint_zju, zhaozhou, wufei}@zju.edu.cn
shijx12@gmail.com, shuicheng.yan@kunlun-inc.com,
hanwangzhang@ntu.edu.sg

## Abstract

We propose a novel method, **TwinAct**, to tackle the challenge of decoupling actions and actors in order to customize the text-guided diffusion models (TGDMs) for few-shot action image generation. TwinAct addresses the limitations of existing methods that struggle to decouple actions from other semantics (*e.g.*, the actor's appearance) due to the lack of an effective inductive bias with few exemplar images. Our approach introduces a common action space, which is a textual embedding space focused solely on actions, enabling precise customization without actor-related details. Specifically, TwinAct involves three key steps: 1) Building common action space based on a set of representative action phrases; 2) Imitating the customized action within the action space; and 3) Generating highly adaptable customized action images in diverse contexts with action similarity loss. To comprehensively evaluate TwinAct, we construct a novel benchmark, which provides sample images with various forms of actions. Extensive experiments demonstrate TwinAct's superiority in generating accurate, context-independent customized actions while maintaining the identity consistency of different subjects, including animals, humans, and even customized actors. Project page: https://twinact-official.github.io/TwinAct/.

## 1 Introduction

We are interested in customizing Text-Guided Diffusion Models (TGDMs) (*e.g.*, Stable Diffusion [21]) to generate customized actions specified by a few user-provided images. For example, given a few images of *Black Widow*'s signature three-point landing action, we can represent this action as a unique token $V^*$ and generate imaginary creation by the prompt like *"Leonardo $V^*$"* (the third row in Figure 1). One might question why we do not condition TGDMs on precise textual descriptions [21] or sketch images [31] to generate the desired action. The reason is that actions are often unutterable, and even with finely detailed descriptions (SD column in Figure 1) or skeleton images (ControlNet column in Figure 1), TGDMs cannot accurately follow these instructions.

Further, can we use existing few-shot customized TGDMs [4, 7, 8, 10, 14, 22, 27] to customize actions? Unfortunately, the answer is still no. This is because these methods fine-tune the textual embeddings of $V^*$ in prompts like "*Black Widow $V^*$*" by reconstructing the corresponding image. But, except for actions, there are many other semantics in the exemplar images that are coupled in $V^*$, like the characteristics of the specific actors. Thus, the few-shot action images in those methods

---

*Equal Contribution.
† Corresponding Author.

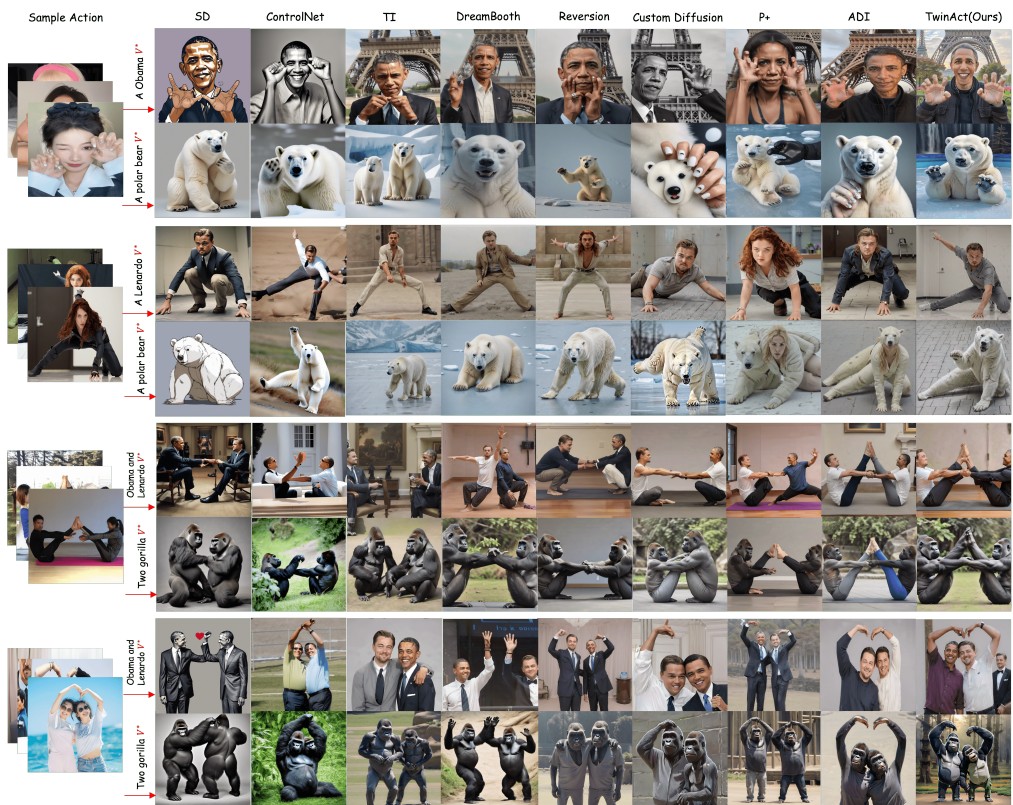

Figure 1: **Qualitative comparisons of TwinAct with other methods.** TwinAct preserves the identity consistency of actors while allowing customized actions to be accurately generalized across different actors by effectively decoupling actions and actors.

lack an effective inductive bias to achieve decoupled action tokens $V^*$ from actors, which results in action-irrelevant information like *Black Widow*'s red curly hair being confounded in the generated images as shown in Figure 1. Although some methods mitigate the effects of missing inductive bias through contrastive learning on textual embeddings or visual images, they still struggle to exclude all action-irrelevant semantics. For example, [8] uses prepositions as positive samples and randomly selects negative samples from a limited set of POS words (*i.e.*, nouns, adjectives). However, this approach can not guarantee the exclusion of all action-irrelevant information from exemplar images for $V^*$, resulting in the inclusion of irrelevant semantics like *manicure* and *red curl hair* (the first and third rows in Figure 1) confounding the customized tokens. On the other hand, [7] and [14] utilize contrastive training data involving the same subject with and without specific customized concepts (*e.g,* melted and closed eyes, etc.) to decouple the subject and customized concepts. However, the creation of extensive contextual training data is costly, particularly when dealing with customized actions that involve multiple subjects.

By observing the real-world process of a director instructing an actor to perform a specific action, such as the signature action of the character *Black Widow* which involves a set of action phrases including "left hand on the ground", "right hand raised behind the back", "left leg on the ground", and "right leg straight out", we find that these phrases concentrate solely on actions without providing any specific details about the actor involved. Moreover, these phrases can be modified or combined to create new actions, as illustrated in Figure 2. Inspired by these observations, we propose a method called **TwinAct**, which aims to decouple the action from the actor by constructing an effective *Common Action Space*. This space is exclusively for actions, eliminating any confusion with actor-specific characteristics and enabling the generation of diverse and creative actions that can be applied across various actors and scenarios.

Our approach involves three steps. **1)** We collect 832 action phrases that cover a range of movements from finger gestures to full body poses, and extract the corresponding textual embeddings using

the tokenizer of CLIP [18]. These textual embeddings are used to build a common action space (Section 3.2) through Principal Component Analysis (PCA [9]); **2)** We employ a Multilayer Perceptron (MLP) to modulate the PCA coefficients to imitate the customized actions within the action space (Section 3.3); and **3)** We introduce an action similarity loss to provide semantic similarity between the exemplar images and generated images, resulting in synthesizing highly adaptable customized action images in diverse contexts (Section 3.3). Extensive experiments demonstrate the superiority of our method over the existing method. Finally, the main contributions of this paper are summarized as:

- We propose **TwinAct** to generate customized action images, which can synthesize novel renditions of user-specific action in different contexts including animals, humans, and even customized actors.
- We introduce a common action space that focuses solely on the action itself, without any actor-specific details. By combining action bases in this space for customized action imitation, we are able to effectively decouple action from actor, allowing for greater flexibility in customization.
- Through extensive experiments, we demonstrate that TwinAct outperforms previous methods in maintaining the fidelity of customized actions while also preserving the consistency of actor identities.

## 2   Related Work

### 2.1   Image Generation and Customization

The text-guided diffusion models (TGDMs) [15, 19, 20, 21, 23] have emerged as a powerful paradigm in the domain of image generation, enabling the creation of image variations that are aligned with specific textual descriptions. Recently, the customized TGDMs [4, 5, 8, 10, 13, 11, 22] aim to encoder the customized concept within the textual embedding, often represented by a unique token and decode the token into pixels by the cross-attention in the U-net decoder [24]. Textual Inversion [4] optimizes textual embedding and synthesizes personalized images by integrating the concept token with the target prompt. DreamBooth [22] extends this concept by proposing a framework that optimizes all parameters of the denoising U-Net architecture, based on a specific token and the class category of the subject. Several other works [5, 10] have focused on optimizing subsets of weights or introducing additional adapters to achieve more efficient optimization and better conditioning of the generated images. For instance, Custom Diffusion [10] fine-tunes only the cross-attention layers within the U-Net, while P+ [27] expands the textual-conditioning space with per-layer tokens to allow for greater disentanglement and control over the generation process. Despite these methods achieving commendable results in customized actors, they struggle with generating customized action images, as shown in Figure 1, due to the lack of an effective inductive bias with few exemplar images.

### 2.2   Customized Action Image generation

In contrast to the rapid progress in customized actors, customized action has received less attention in the community. Since the complex details such as specific angles, body positions, and motion trajectories, describing a specific action is more challenging than describing a specific actor. Even with finely detailed descriptions (see Appendix A.1) TGDMs still cannot accurately follow these textual descriptions. A straightforward approach to generate a specific action image is ControlNet [31] which is conditional on a given skeleton image to generate images. However, they only provide a rough approximation of action consistency (see Appendix A.3). Particularly, when it comes to capturing fine details such as fingers, skeleton images may lack the necessary detail to accurately reproduce these actions, leading to noticeable visual defects in the generated images. In addition, when conditioned on detailed images (see Appendix A.3), they suffer from limited diversity and flexibility. Recent advancements for customized TGDMs preliminary explore action-based customization. Reversion [8] proposes relation-steering contrastive learning that aims to guide the relation prompt towards relation-dense regions within the text embedding space, thereby disentangling the learned relation from other concepts like appearances. Lego [14] and ADI [7] construct contexts involving the same subject with and without a specific concept, to decouple the subject and specific concepts. However, these methods by employing contrastive learning on textual embeddings or visual images can not effectively decouple actions from actors. This is because they cannot exclude all irrelevant concepts with a limited number of negative samples.

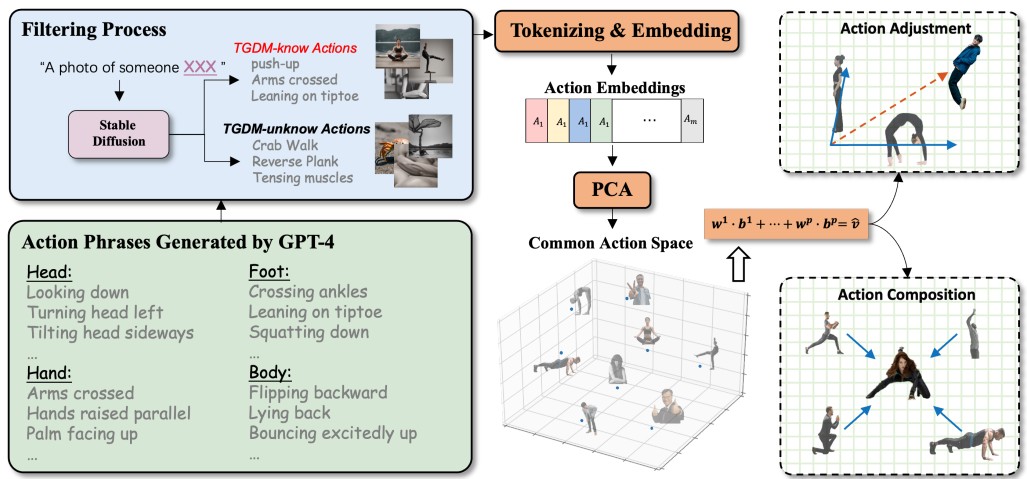

Figure 2: **The construction of Common Action Space.**

# 3 Methods

Imagine the following scenario: When the character *Black Widow* performs her signature three-point landing action, the director may give the following instruction "*Bend knees and lower body down into a squatting position; Place right hand forward, and right arm should be fully extended*". We can see that when people are asked to perform a specific action, it is often accompanied by instructions that involve the composition of various body movements and the adjustment of angles and amplitudes as shown in the right part of Figure 2. Inspired by the way in which humans perform specific actions, we propose **TwinAct** which applies similar principles to action customization. Specifically, we first collect a set of basic actions and encode them using a pre-trained text encoder to build a common action space represented by a set of action bases. Subsequently, we imitate the customized action by decomposing it into a combination of action bases with corresponding weights. Finally, we utilize both the reconstruction loss and action similarity loss to fine-tune the customized token embedding in the text encoder and the LoRA [6] layer in the text-to-image decoder to generate high-fidelity images of the customized action.

## 3.1 Preliminaries

Our study is based on the text-guided diffusion models (TGDMs) which consist of a textual encoder $\mathcal{E}$ and a text-to-image decoder $\mathcal{D}$. Given a few sample action images $\mathbf{x}$ and a textual prompt $c$ with a customized token $V^*$, such as "a photo of $V^*$", we aim to fine-tune the embedding of customized token $V^*$ in $\mathcal{E}$ and the LoRA layer in $\mathcal{D}$ to realize action customization. The textual encoder $\mathcal{E}$ first tokenizes the prompt $c$ into a set of token embeddings $\mathbf{h}$, which are then used to generate a textual condition $\tau_\theta(c)$ with the text transformer in $\mathcal{E}$. Subsequently, the textual condition $\tau_\theta(c)$ is utilized by the conditional denoising diffusion model $\epsilon_\theta(\cdot)$ in $\mathcal{D}$ to generate images. The commonly used training objective is:

$$\mathcal{L}_{rec} = \mathbb{E}_{\epsilon,t,\mathbf{x}_0,c}\left[\|\epsilon - \epsilon_\theta(\mathbf{x}_t, t, \tau_\theta(c))\|^2\right] \tag{1}$$

where $t$ is the timestep, $\mathbf{x}_0$ is the original image and $\mathbf{x}_t$ is a noisy image constructed by adding noise $\epsilon \sim \mathcal{N}(0,1)$ to $\mathbf{x}_0$. After training, any textual condition $\tau_\theta(c)$ with $V^*$ can generate customized action in a new context defined by prompt $c$.

## 3.2 Step-1: Building Common Action Space

The initial step involves identifying the set of action bases that defines the common action space within the text-to-image diffusion model.

First, we task GPT-4 [1] with generating a range of common action phrases based on different body parts, including head, fingers, hands, and full body (see Appendix B.2). Then, we carefully sift through the generated action phrases to eliminate duplicates. Subsequently, we build a manual filter based on the pre-trained text-to-image diffusion model by constructing prompts and synthetic images

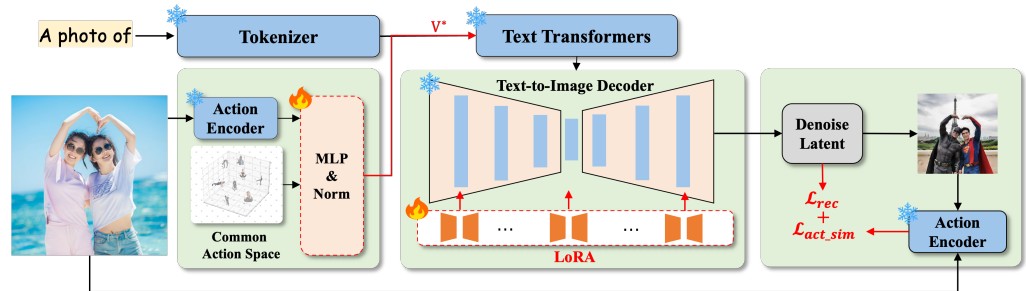

Figure 3: **The overview of the proposed TwinAct.** We optimize the coefficients of the action bases to avoid encoding the action-irrelevant features. After training, we combine the learned coefficients and shared action base to generate images with the customized action.

of each action phrase to remove those action phrases that are unknown to the text-to-image diffusion model, *e.g.,* "Reverse Plank" as shown in Figure 2.

After the filtering process, we identify a total of $m = 832$ action phrases. Each phrase $a_i$, where $i \in \{1, ..., m\}$, is tokenized and encoded into an action embedding group $A_i = [a_1^i, ..., a_k^i]$. It is important to note that the length $k$ of each action embedding group $A_i$ varies, as each action may consist of a different number of words. Through empirical observation, it was found that action phrases with three words or fewer are the most common, prompting the designation of each $A_i$ as containing three embeddings (*i.e.*, $k = 3$ for all $m$ actions). For simplicity, we denote the three embeddings of all $A_i$ as $Q_1$, $Q_2$, and $Q_3$ respectively, where $Q_k = [a_k^1, ..., a_k^m]$.

Inspiring by prior research [2] employing PCA to transform high-dimensional facial data into a lower-dimensional space, we calculate the PCA transformation matrix $B_k$ for each set $VQ_k$ using the function $\text{PCA}(Q_k, p)$. This transformation plays a crucial role in reducing redundant information within the data and enhancing the efficiency of model optimization. Specifically, $\text{PCA}(Q_k, p)$ reduce dimensionality of $Q_k \in \mathbb{R}^{m \times d}$ and generate $p$ principal components as $B_k = [b_k^1, ..., b_k^p] \in \mathbb{R}^{p \times p}$, which captures the essential features of data in a more compact form. Our experimental findings indicate that optimal results are achieved when $p = 512$ (see Section 4.3). Finally, we obtain a set of action bases as $[B_1, B_2, B_3]$, which define an action space that does not contain any action-irrelevant information thus providing a strong inductive bias to achieve decoupled action token embeddings.

### 3.3  Step-2: Imitating Action via Action Bases

This step involves searching for the optimal parameters to combine the action bases for imitating customized actions.

Our experiments (see Figure 1) have shown that existing methods, which aim to identify optimal personalized token embeddings within the vast text space of pre-trained text encoders, struggle with the inclusion of non-action information. And even with contrastive learning, they are still unable to exclude all irrelevant information. Fortunately, action bases offer an alternative approach for optimizing the customized token embedding. By combining action bases $[B_1, B_2, B_3]$ with corresponding coefficients $[W_1, W_2, W_3]$, where $W_k = [w_k^1, ..., w_k^p]$, new actions can be imitated as shown in Figure 2. To achieve this, we first utilize a Multi-Layered Perceptron (MLP) to combine the action bases. While backpropagation is a common method for determining optimal coefficients, our experiments indicate its limited effectiveness due to the scarcity of user-provided images (see Section 4.3). Therefore, we use CLIP as an action encoder to extract semantic features as a prior. Specifically, given an action sample image $\mathbf{x}$, we first extract the action features $\mathbf{x}_p$ with action encoder $\psi(\mathbf{x})$, then MLP is used to map $\mathbf{x}_p$ into the modulating coefficients $[W_1, W_2, W_3]$ as:

$$W_k = \text{MLP}(\mathbf{x}_p) \tag{2}$$

Finally, for a customized action, we can combine the action bases to imitate the action as:

$$\hat{V} = [\hat{v}_1, \hat{v}_2, \hat{v}_3], \hat{v}_k = \sum_{j=1}^{p} w_k^j b_k^j \tag{3}$$

where $\hat{V}$ is the embedding of the customized tokens $V^*$. In this way, $\hat{V}$ is a linear combination of the action bases. Consequently, the vector interpolated in the action space would not contain any action-irrelevant features, allowing us to achieve the objective of decoupling the action and the actor.

### 3.4 Step-3: Generating Customized Action

The final step involves optimizing the embeddings of the customized token $V^*$, which is derived from the action base through linear combinations and synthesis customized action image.

**Training.** Initially, a few user-provided images and corresponding textual prompts containing $V^*$ are used to apply the pixel reconstruction-based loss $\mathcal{L}_{rec}$, similar to previous customized TGDMs (in Eq 1). However, the traditional $\mathcal{L}_{rec}$ in existing TGDMs aims to faithfully reconstruct the sample image appearance, focusing on low-level pixel features. This narrow focus limits the model's ability to recognize and invert high-level action features in the images, as shown in Figure 7(b).

To address this limitation, we introduce an action similarity loss to enhance the model's grasp of high-level action semantics. Beginning with an input image $\mathbf{x}$ and a corresponding prompt $c$, we randomly initialize a latent variable $\mathbf{x}_T$. This variable is progressively denoised until it reaches a randomly selected timestep $t$, at which point the denoised image $\mathbf{x}^{prd}$ is predicted directly from $\mathbf{x}_t$. The goal is to evaluate the action similarity between these two images. Specifically, by leveraging action encoder $\psi\left(\cdot\right)$, the encoded action features for both the reference action $\mathbf{x}_p$ and the generated action $\mathbf{x}_p^{prd}$ are extracted. We then calculate the cosine similarity between these action features to measure action consistency during the generation process. This similarity is formally defined as:

$$\mathcal{L}_{act\_sim} = \mathbb{E}_{c\sim p(c)}\mathbb{E}_{x\sim p(x|c)}\left[1 - \cos\_\text{sim}(\mathbf{x}_p, \mathbf{x}_p^{prd})\right] \tag{4}$$

By integrating the action similarity loss, the model's focus shifts from detailed pixel information to high-level action semantics. The final training loss for TwinAct can be expressed as:

$$\mathcal{L} = \mathcal{L}_{rec} + \mathcal{L}_{act\_sim} \tag{5}$$

**Testing.** After training, the three groups of coefficients $[W_1, W_2, W_3]$ and the LoRA parameters in the text-to-image decoder are saved. Users can generate images depicting customized actions in different contexts by providing prompts such as *"Leonardo $V^*$"*.

## 4 Experiment

### 4.1 Experiment Setup

**Dataset.** We tackle the challenge of decoupling specific actions from the actors in user-provided images. Since there are no publicly available customized action datasets, we introduce a novel benchmark, consisting of 12 actions involving multiple body parts, such as fingers, arms, legs, and full-body motions. Each action contains approximately 10 sample images. In addition to actions performed by a single actor, we incorporate more complex actions that involve multiple actors.

**Metrics.** To comprehensively validate the efficacy of TwinAct, we incorporate objective metrics to assess the quality of the generated images. We calculate the consistency between the action in the sample image and the generated image through the CLIP score, which is denoted as "$S_{Action}$". Additionally, ensuring identity consistency is a crucial aspect of our task, so we evaluate the actor's identity similarity using a pre-trained face recognition encoder [3], which is denoted as "$S_{Actor}$". In addition, we also conduct user studies including "$U_{Action}$" and "$U_{Actor}$", representing user preferences for action fidelity and actor identity consistency in the generated images.

### 4.2 Comparison with State-of-the-art Methods

**Qualitative Comparison.** We illustrate the advancements of our method compared to previous approaches in handling various forms of actions, including hand, body, single-actor, and multi-actor actions, as shown in Figure 1. Initially, we evaluate two methods for generating customized action images without fine-tuning, as shown in the column of SD and ControlNet in Figure 1. Images generated from detailed textual prompts (SD) frequently exhibit misalignment with the actions in the sample images. On the other hand, Controlled Generative Models (ControlNet), which depend on

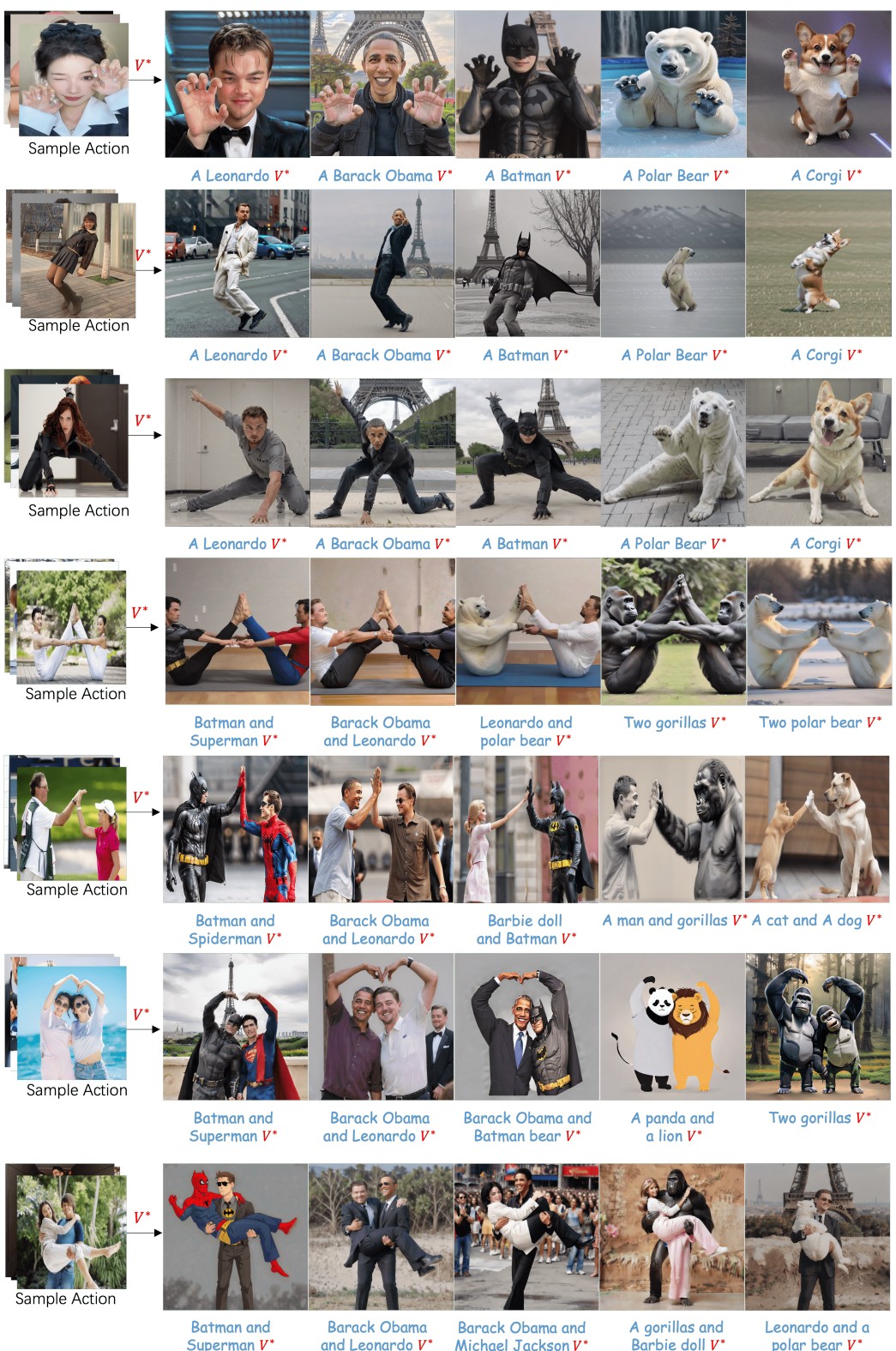

Figure 4: **The results of customized action generation with TwinAct.** TwinAct generates images of different actors performing customized actions such as celebrities and animals, and maintains the consistency of the action and identity of the subject.

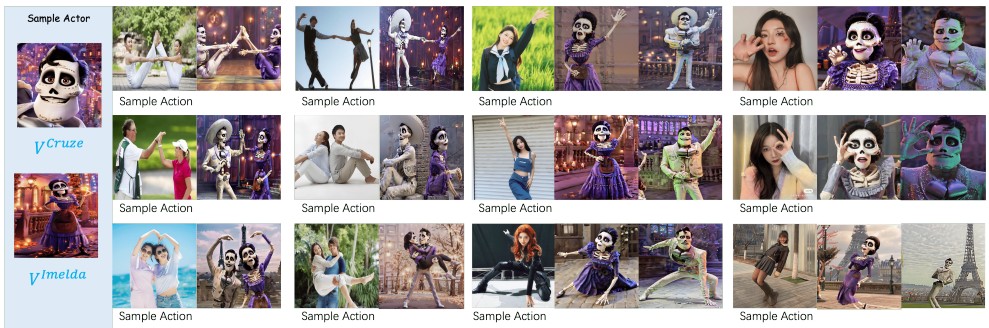

Figure 5: The results of **customized actors** performing **customized actions** generated by TwinAct.

| Methods | | Text Inversion | DreamBooth | Reversion | Custom Diffusion | P+ | ADI | Ours |
|---|---|---|---|---|---|---|---|---|
| Objective | $S_{Action}$ | 9.12 | 12.23 | 18.73. | 26.83 | 33.95 | 45.32 | **69.47** |
| | $S_{Actor}$ | 33.85 | 38.53 | 45.58 | 41.58 | 46.12 | 48.76 | **73.34** |
| User | $U_{Action}$ | 0.22 | 0.19 | 0.89 | 1.16 | 1.30 | 2.25 | **3.86** |
| Study | $U_{Actor}$ | 0.53 | 0.70 | 1.05 | 1.31 | 1.23 | 2.08 | **4.12** |

Table 1: **Quantitative comparisons.** We evaluate TwinAct based on both objective metrics and user study. The results indicate that TwinAct surpasses all baseline methods.

skeleton images, tend to only imitate the basic structures of the sample images and lack a profound understanding of the contextual dependencies required for action customization. This deficiency leads to significant body deformations and flaws, particularly when animals are involved in the action. For example, in the fifth and sixth rows of Figure 1, both the human and the gorilla mistakenly raise their hands instead of their feet.

Subsequently, we conduct comparisons with state-of-the-art baselines such as Textual Inversion [4], DreamBooth [22], Reversion [8], Custom Diffusion [10], P+ [27], and ADI [7]. Textual Inversion and DreamBooth tend to capture minimal action-related information, primarily because of their limited tunable parameters. On the other hand, Custom Diffusion, which fine-tunes cross-attention in U-net, and P+, which utilizes per-layer token embeddings, manage to retain more information from the sample images. However, they uncontrollably invert action-irrelevant details in the sample images and bring these details into the generated image, such as the human hand and the manicure in the first and second rows shown in Figure 1. In addition, Reversion and ADI try to exclude action-irrelevant information from the images by contrastive learning but face challenges in comparing all negative samples. Therefore, the images they generate still contain some confounded details, such as the red curly hair of the *Black Widow* in the third row and the yoga pants in the sixth row. In contrast, TwinAct shows the best customized action generation results. Through the acquisition of a customized action token embedding, which is derived from a linear combination of action bases within the common action space, TwinAct accurately capture the action information in the sample images while excluding irrelevant details with good generalization.

**Quantitative Comparison.** In addition to the visual quality, we also present a numerical comparison between TwinAct and the baselines in Table 1. As can be seen, our approach outperforms in terms of both action and actor similarity, demonstrating its effectiveness in imitating customized actions and filtering out irrelevant information that could potentially confuse the portrayal of the new actor. Furthermore, the performance across four representative types of action customization tasks (*i.e.*, Single-Actor, Two-Actor, With-Human, With-Animal) is illustrated in Figure 6. Our method consistently shows superior performance and the least variability across all types. Notably, the method labeled as P+ experiences the most significant performance drop when involving animals, and ADI exhibits a considerable performance drop when handling actions involving two actors.

Additionally, following the methodology outlined in [30], we conduct a quantitative comparison involving human evaluators to further validate the effectiveness of TwinAct. We invite 100 users to rate each pair of actor-action images generated by TwinAct and baselines on a scale ranging from 1 (worst) to 5 (best), focusing on both action and actor consistency. The collective feedback, as presented in Table 1, clearly indicates a strong preference for TwinAct among the users.

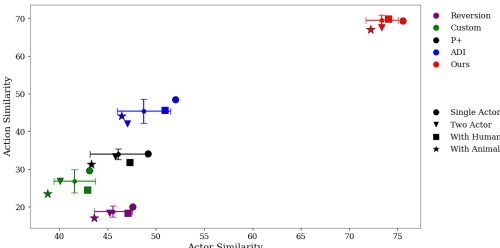

Figure 6: Numerical analysis in terms of the actor and action similarity on four prompt types.

Table 2: Ablation studies.

| Methods | $S_{Actor}$ | $S_{Action}$ |
|---|---|---|
| w/o action space | 58.86 | 52.18 |
| w/ 50% action phrases | 68.39 | 61.86 |
| w/o action encoder | 67.27 | 59.06 |
| w/o $\mathcal{L}_{act}$ | 70.23 | 63.58 |
| $p = 64$ | 72.47 | 54.27 |
| $p = 256$ | 73.12 | 61.66 |
| $p = 768$ | 71.62 | 67.63 |
| Ours | **73.34** | **69.47** |

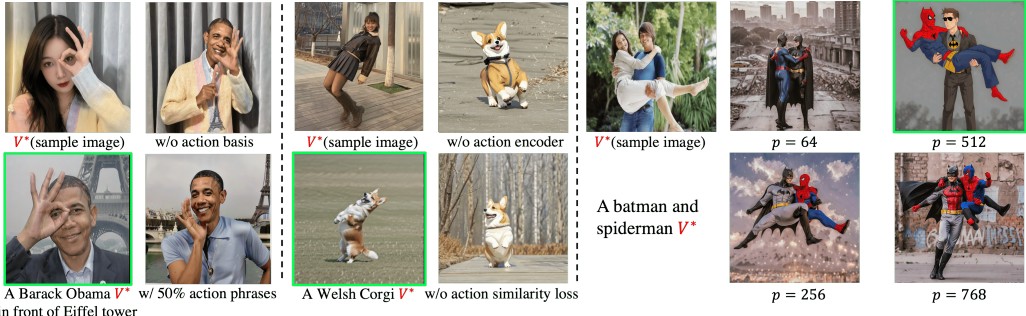

Figure 7: **Visualization of ablation studies.** The results of TwinAct are highlighted in green box.

## 4.3 Ablation Study

**Impact of Action Space.** We assess the impact of the number of action phrases for building the common action space, as shown in Figure 7(a). In extreme cases, if there exists no action space and we directly learn the customized token embedding from the action encoder (w/o action space), the model can not generate customized actions and is overfitted to the input image details, such as the actor's clothes and the background. When utilizing a smaller number of action phrases (w/ 50% action phrases), the generated quality is not as good as TwinAct.

**Impact of Action Encoder and Action Similarity Loss.** We assess the impact of the action encoder and action similarity loss in Figure 7(b) and Table 2. The experimental results reveal that employing an action encoder can enhance discriminative capabilities regarding actions and yield satisfactory outcomes. In contrast, without an action encoder, optimizing the coefficients $W_k$ of the action bases $B_k$ through back-propagation (w/o action encoder) is ineffective. Furthermore, we can find action similarity loss can guide model towards more accurately imitating the customized actions by drifting the model's focus from low-level pixel features to action-related high-level semantic features.

**Impact of the Number of Principal Components.** We explore the effect of the number of principal components $p$ on model performance. With 768 dimensions in the CLIP text embeddings, we conduct experiments with different values of $p$ from $\{64, 256, 512, 768\}$. The experimental results, outlined in Table 2, indicate that the optimal results are achieved when $p$ is set to 512. Additionally, we showcase the variations in the generated images for different $p$ values in Figure 7(c).

## 4.4 Combinations of Customized Actors and Customized Actions

Our method demonstrates strong adaptability to integrate with existing subject-driven customized generation methods[11], as shown in Figure 5. It effectively adapts to customized actors, which were unknown by TGDMs, and generates visually engaging images showcasing these actors performing customized actions. In contrast, as depicted in Figure 10 in Appendix, existing methods struggle to accurately capture action-related features, resulting in unsatisfactory generation of customized actions and a lack of identity consistency for the customized actors.

# 5    Conclusion

In this paper, we introduce a novel approach, TwinAct, for customized action image generation, which aims to preserve the fidelity of the action and the consistency of the actor's identity. TwinAct consists of a common action space and can create new actions by adjusting or combining the action bases in the space. By imitating action via action bases, TwinAct effectively decouples action from the actor. Extensive experiments demonstrate TwinAct's superiority in generating accurate, context-independent customized actions across various subjects, including humans, animals, and customized actors. Furthermore, TwinAct's ability to preserve the high fidelity and identity consistency in generated images highlights its robustness and adaptability. The potential applications of TwinAct in areas such as animation, gaming, and visual effects are vast, offering new opportunities for creative and personalized content generation.

# 6    Acknowledgements

This work was supported by the National Natural Science Foundation of China (No.62307032, No.62037001) and the Key Research and Development Program of Zhejiang Province (No. 2023C03192).

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

In this appendix, we first provide an in-depth analysis of Stable Diffusionin in Section A.1 and ControlNet in Section A.3, and additional results of TwinAct in Section A.3 and A.4, demonstrating its effectiveness in generating customized action images. Then we introduce the evaluation setting in Section B.1, the details of the action space in Section B.2, and the implementation details in Section C. Finally, we discuss the limitations in Section D and societal implications in Section E for TwinAct.

# A More Qualitative Results

## A.1 Comparison with Stable Diffusion

We first utilize GPT-4 to generate a detailed description of the actions in the sample image provided by the user, as shown in Figure 12. We can find that on the one hand, although GPT-4's description is highly sufficiently, it still inevitably loses many details of the action. On the other hand, existing text-guided image diffusion generation models have difficulty in understanding such detailed textual descriptions, like "heart" in the fourth row in Figure 12. Both reasons result in the generation of action images that are far from the sample image. In contrast, our approach accurately captures action-related features, and the user only needs to provide textual prompts like "a polar bear $V^*$", and then customized action images can be generated in the desired contexts.

## A.2 Comparison with ControlNet

In addition to controlling the generation of models through textual conditioning, current work such as ControlNet supports extracting spatial constraints from reference images as conditions to generate images. We tried two reference images, skeleton images, and sketches, as conditions to control the generative model to generate customized action images. The results are shown in Figure 9. We can find that the skeleton image-based approach loses a lot of details of the action, such as fingers, and it is difficult to imitate the customized action. The other sketch-based approach provides more details, but this limits the diversity of the model generation results, e.g., it is difficult to generate images of animals performing customized actions since the sketches contain human bodies. Also, the sketches contain some additional action-irrelevant information such as the background, the actor's hairstyle, and so on. More importantly, the controlled generation model lacks the understanding of the contextual dependency of the action, as shown in the third row in Figure 9. It is difficult for the generation model to understand the relationship between the hands and the feet in the customized action, which leads to serious body distortions and visual deficiencies, and this becomes even more obvious when animals are involved.

## A.3 Comparison with Baseline on Customized Actor

In this section, we compare the results of existing customized TGDMs and TwinAct in generating images containing customized actors and customized actions. As shown in Figure 10, our approach significantly outperforms the previous methods in maintaining the fidelity of the action and consistency of the character's identity. Textual Inversion, Dreambooth, and Revrsion struggle to imitate the customized action. As for, Custom Diffusion, P+, and ADI, although they can roughly maintain the consistency of the actions specified in the sample image, fail to maintain the identity information of the customized actors. This is because their customized action tokens are confounded with information irrelevant to the actions in the sample image, *e.g., Black Widow*'s red curly hair in the first row, and the yoga pants in the second row.

## A.4 Additional Results

In this section, we show more customized action images generated by TwinAct, as shown in Figure11, covering the rest of the 5 actions within our benchmark.

# B Experiment Design

## B.1 Evaluation Setting

Following [7], we provide 25 subjects for evaluation as follows:

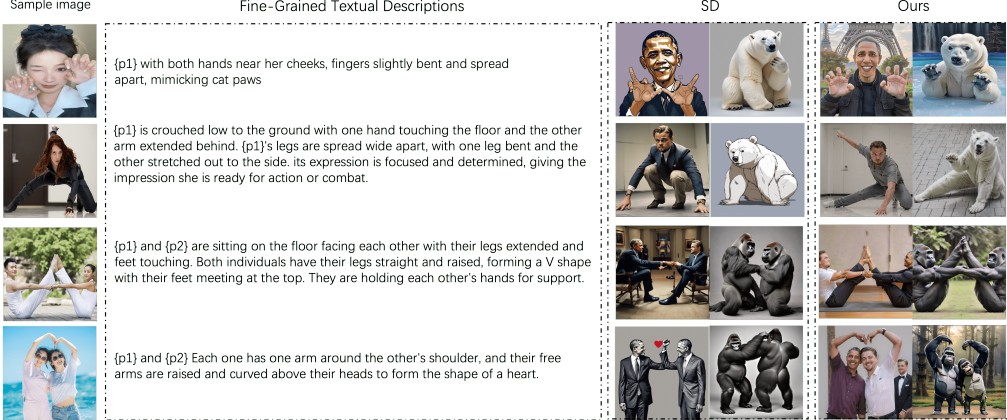

Figure 8: **Comparing the results generated by TwinAct and fine-grained textual descriptions.** The results show that customized actions are difficult to describe and that existing text-to-image generation models do not accurately follow textual instructions even when fine-grained textual descriptions are provided.

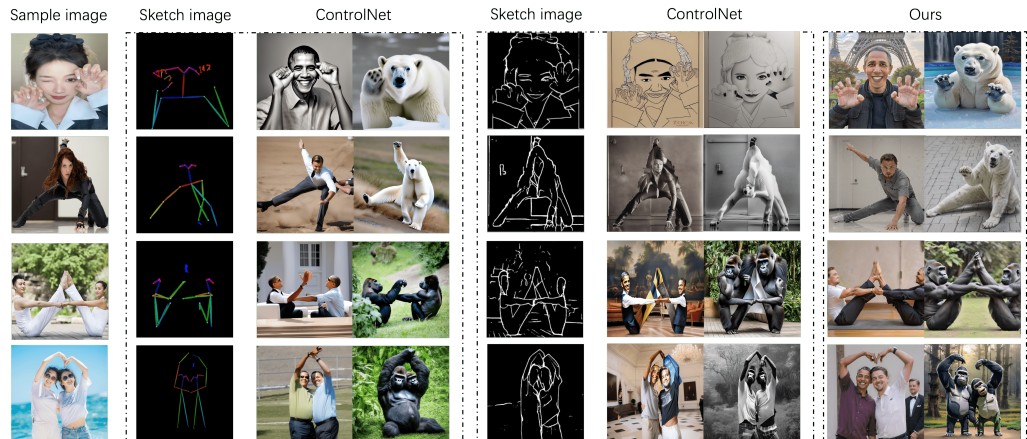

Figure 9: **Comparison of results generated by TwinAct and sketch-based generation models.** The results show that it is difficult for the user to provide a suitable sketch for generating customized action images. The results generated with skeleton images show it is difficult to capture the details of the action such as fingers, while the results generated with line images are limited in generalization, especially when it involves animals.

- **generic human:** "A boy", "A girl", "A man", "A woman", "An old man"
- **well-known personalities:** "Barack Obama", "Michael Jackson", "David Beckham", "Leonardo DiCaprio", "Messi", "Spiderman", "Batman"
- **animals:** "A dog", "A cat", "A lion", "A tiger", "A bear", "A polar bear", "A fox", "A cheetah", "A monkey", "A gorilla", "A panda"
- **customized actors:** "Cruze", "Imelda"

where the inclusion of diverse and previously unseen actors, as well as animals and customized characters, requires models to retain pre-trained knowledge while also generating accurate and undistorted representations of these actors.

## B.2    Action phrases for Action Sapce

As stated in the manuscript, we devise a set of action phrases to build action space. Specifically, we collect a comprehensive list of about 1,200 action phrases with the help of GPT-4 [1]. After obtaining these action phrases, in order to filter out those action phrases that are not recognized by the TGDMs.

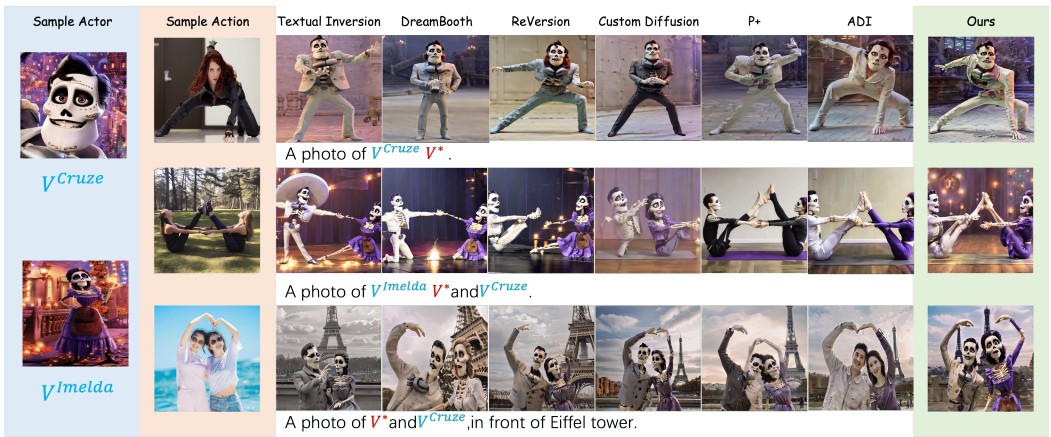

Figure 10: **Comparing the results of generating customized actors to perform customized actions generated by TwinAct and other methods.** TwinAct can better maintain the identity consistency of the subject in the generated image, and the fidelity of the action.

we synthesize the image result by providing 2 types of prompts, *i.e.* "A photo of *{Action}*", "A photo of someone *{Action}*", to the Stable Diffusion XL [17], where *{Action}* represents an action phrases (*e.g.* "raise hand"). Those action phrases that do not generate the correct action, such as *"reverse plank"*, or *"crab walk"*, are filtered out. We ended up with 832 valid common action phrases, which were then further tokenized and encoded into the action space.

## C More Implementation Deatils

### C.1 Implementation Details about TwinAct

**Implementation Details.** For our method, we use the AdamW[12] optimizer with a learning rate of 2e-4. We use CLIP as a preprocessor to estimate the action of the given reference image. Unless otherwise specified, Stable Diffusion XL is selected as the default pre-trained model, and images are generated at a resolution of 1024×1024. All experiments are conducted on A-100 GPUs. We fine-tuning text embedding along with the LoRA layer. We integrate the LoRA layer into the linear layer within all attention modules of the U-net, utilizing a rank of $r = 8$.

**Sample Details.** All experiments and evaluations make use of the DDPM [25] with 50 sampling steps with a scale of 7.5 for all methods. To ensure consistency and filter out undesired variations in diffusion models, we follow the approach outlined in [11] by employing the same negative prompt for both our method and the comparison methods during sampling. The negative prompt used is "long body, low-res, bad anatomy, bad hands, missing fingers, extra digit, fewer digits, cropped, worst quality, low quality."

**Running Times.** The process of tuning a customized action token embedding in the text encoder and the LoRA layer in the text-to-image decoder typically requires approximately 15 minutes using an Nvidia-A100 GPU.

### C.2 Implementation TwinAct with Customized Actor

Thanks to the rapid development of subject-driven customized text-guided diffusion models for image generation, they achieved an exciting proceeding in generating high-fidelity customized actor images. We combine TwinAct with the existing state-of-the-art subject-driven customized TGDMs [11] to generate images of customized actors performing customized actions. To achieve this, we only need to apply the action base alone to the embedding of the customized action tokens, while the embedding of the customized actor's tokens is obtained as in [11]. The action similarity loss can be applied to the optimization of both the customized action tokens and the customized actor tokens, and our generation results show that the action similarity loss does not affect the learning of the customized actor tokens.

# D   Limitation and Future Work

In this work, our goal is to generate images of customized actions. There are two limitations to TwinAct. First, TwinAct obtains the embedding of customized action toknes by combining action bases. Although the action base is carefully defined, it is still possible to miss some actions, which limit the expressiveness of TwinAct's output domain. This can be overcome by further expanding the action space by adding more common action phrases. Another limitation comes from the physical defects of TGDMs in generating human bodies, we mitigate this problem by providing some negative prompts like "long body, low-res, bad anatomy, bad hands, missing fingers, extra digit, fewer digits, cropped, worst quality, low quality". Furthermore, there are some works [29, 16, 28] designed to solve the problem. Moreover, TwinAct is not restricted to image generation models; it can also be used for customizing text-guided diffusion models to generate videos or 3D assets. we leave it as future work.

# E   Expected Societal Implications

We aim to address the challenge of decoupling actions and actors to customize the text-guided diffusion model for few-shot action image generation. A primary ethical concern is the potential misuse of this technology, notably in creating deepfakes, which can result in misinformation, privacy violations, and other harmful outcomes. To mitigate these risks, the establishment of robust ethical guidelines and continuous monitoring is essential.

The concern raised here is a common one, not just for our method but across various multi-concept customization techniques. A viable strategy to lessen these risks might be the implementation of tactics akin to those used in anti-dreambooth[26]. This approach involves adding minor noise disturbances to the shared images, thereby hindering the customization process. Furthermore, embedding invisible watermarks in the generated images can serve as a deterrent against misuse and ensure that they are not used without due acknowledgment.

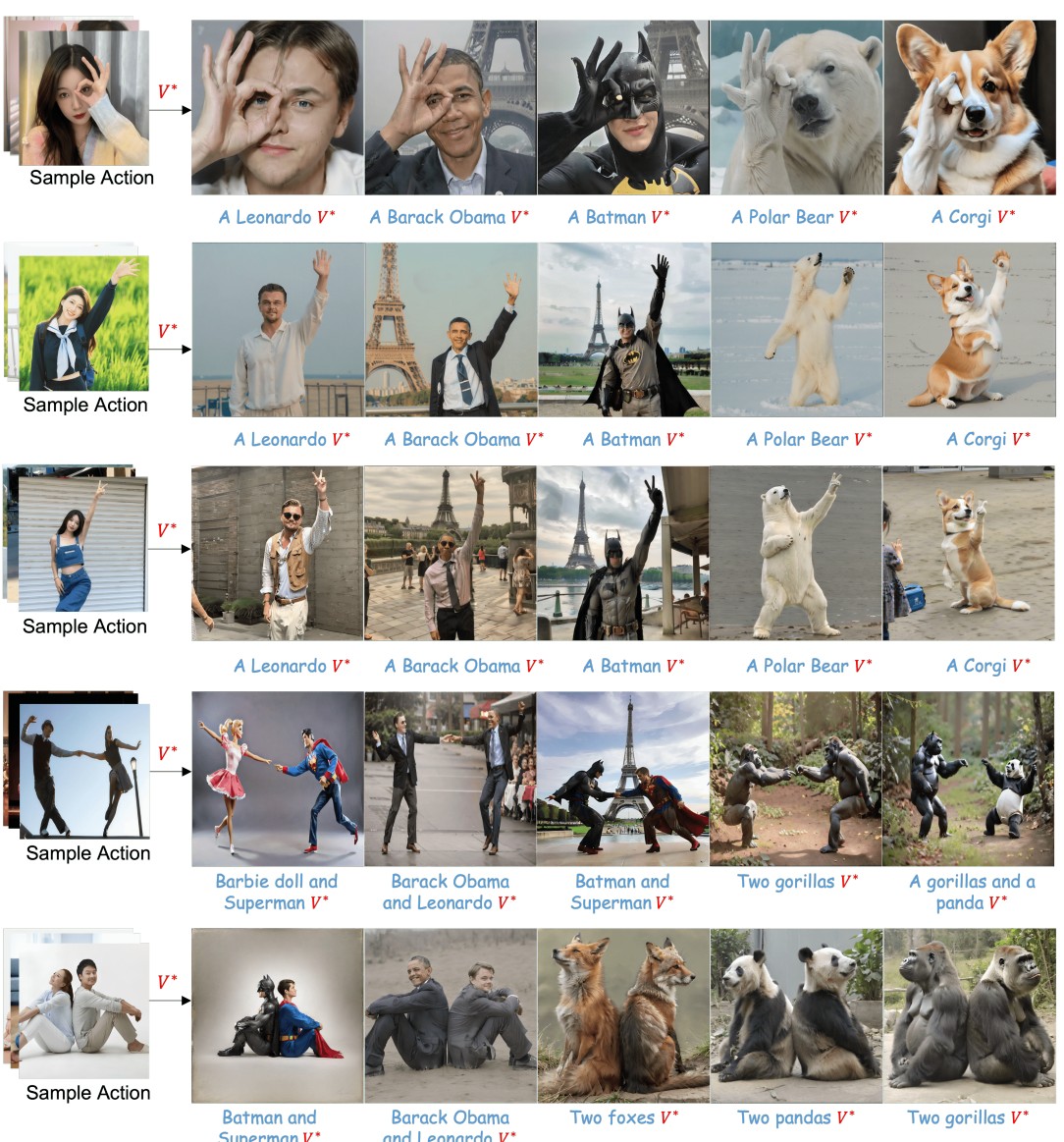

Figure 11: **More TwinAct generated results.**

| Hands | Finger | Heads | Full-Body |
|---|---|---|---|
| Hand on shoulder | Fingers crossed | Nodding head yes | Crouching stealthily |
| Waving goodbye | Finger snapping | Shaking head no | Sprawling out |
| Slapping forehead | Thumbs up | Tilting head sideways | Pouncing forward |
| Covering mouth | Spinning finger | Looking up | Embracing warmly |
| Hand on heart | Finger gun | Looking down | Shoving forcefully |
| Casting spell | Tapping fingers | Turning head left | Expanding chest |
| Raising hand | Fist bump | Turning head right | Contracting abdomen |
| Folding arms | Palm up asking | Tucking chin | Twirling gracefully |
| Crossing arms | Praying hands | Raising eyebrows | Pirouetting steadily |
| Pointing forward | Knuckle cracking | Furrowing brows | Bearing down |
| Punching air | Hand on shoulder | Blinking eyes | Leaping high up |
| Hugging self | Peace sign | Winking eye | Ducking quickly down |
| Pushing away | Pinching gesture | Squinting eyes | Tumbling forward roll |
| Pulling rope | Wrist flex | Gazing forward | Flipping backward somersault |
| Lifting weights | Slapping forehead | Glancing sideways | Shielding face hands |
| Clapping hands | Grabbing wrist | Staring intently | Kneeling on ground |
| Shaking hands | Finger circling | Glaring angrily | Squatting to rest |
| Patting back | Clenched fist | Smiling broadly | Push-up |
| Swinging arm | Open palm | Frowning deeply | Plank |
| Stretching arm | Pointing down | Puckering lips | Sit-up |
| Waving hand | Thumbs down | Biting lip | Crunch |
| Shaking fist | Palm facing down | Pressing lips | Burpee |
| Brushing hair | Finger tips touch | Opening mouth | Squat |
| Scratching head | Holding object | Closing mouth | Lunge |
| Lifting arm | Boxing punch | Yawning widely | Deadlift |
| Grabbing shoulder | Counting fingers | Chewing food | High knees |
| Reaching out | Wagging finger | Gritting teeth | Butt kick |
| Twisting arm | Wiping tears | Pouting lips | Handstand |
| Flexing wrist | Typing gesture | Sticking tongue | Arching body backward |
| Flipping hand | Finger pointing up | Licking lips | Curling body forward |
| Pulling back arm | Finger pointing sideways | Tilting head up | Leg swings |
| Swinging arm forward | Finger pinching | Tilting head down | Leg lifts |
| Swinging arm backward | Thumb flexing | Tilting head left | Leg drops |
| Throwing arm forward | Finger curling | Tilting head right | Bicycle kicks |
| Pulling arm up | Finger pressing | Rotating head clockwise | Flutter kicks |
| Pushing arm down | Thumb pressing | Jerking head back | Crisscross kicks |
| Circling arm | Finger rubbing | Jerking head forward | Scissor kicks |
| Shielding with arm | Finger crossing over | Jerking head sideways | Side kicks |
| Lowering hand | Finger crossing under | Scratching chin | Front kicks |
| Shaking arm | Finger forming heart | Puffing cheeks | Back kicks |
| Resting arm on hip | Finger forming circle | Gnashing teeth | Inward heel taps |
| Leaning on elbow | Thumb pointing left | Stroking chin | Outward heel taps |
| Draping arm over head | Thumb pointing right | Lowering eyelids | Toe touches |
| Tucking arm behind back | Thumb sliding down | Sucking on a tooth | Walking |
| Swinging arm side to side | Thumb sliding up | Pinching nose | Running |
| Lifting arm over head | Finger forming V | Widening eyes | Jogging |
| Lowering arm to side | Make a fist | Narrowing eyes | Sprinting |
| Extending arm forward | Put palms together | Rubbing nose | Skipping |
| Resting arm on knee | Finger snap | Rubbing eyes | Jumping |

Figure 12: **Some examples of action phrases.**

