# OpenReview forum: "Action Imitation in Common Action Space for Customized Action Image Synthesis"
_NeurIPS.cc/2024/Conference — NeurIPS 2024 poster_

### Official Review · Reviewer_2ZgR · 2024-07-07

**Soundness:** 3
**Presentation:** 3
**Contribution:** 3
**Rating:** 7
**Confidence:** 4

**Summary:**

The paper introduces “TwinAct”, a novel method for separating actions from actors in few-shot action image generation using text-guided diffusion models (TGDMs). It creates a “common action space” to focus on actions alone, allowing for precise customization without actor-specific details. The process is streamlined into three main steps: constructing the action space from key phrases, imitating the action within this space, and generating varied, context-adaptive images using an action similarity loss. And the results demonstrate its robustness and versatility for personalized and creative content generation.

**Strengths:**

1.  The idea of imitating actions through a combination of action bases within this space is interesting and provides a new way to achieve few-shot customization.
2.  The paper is well-structured and clearly articulated. The method is technically sound, with a clear explanation of the steps involved, from building the action space to generating the final images.
3.  The experimental results are impressive, especially the robustness of TwinAct. The potential applications for both customized characters and actions are of great interests.

**Weaknesses:**

1.  In Figure 6, why do all the methods show a performance degradation when two people are involved?
2.  Can the authors explain in detail how the model can be customized for both character and action？
3.  Could the authors provide examples and analyses of where TwinAct fails or performs suboptimally? What steps can be taken to address these failure modes?
4.  Adding something like Figures 8 and 9 from the appendix would make the results more convincing for readers who do not venture to the appendix to search for plots.

**Questions:**

Please check the detailed comments in the weaknesses part.

---

> ### Author Rebuttal · Authors · 2024-08-07
>
> ### **For Reviewer** **2ZgR**
>
> We are sincerely grateful to the reviewers for dedicating their time and effort to review our work, and we appreciate the recognition of the novelty of our approach and the significance of our given the impressive result. We will try to address  reviewer's comments in detail below.
>
> ### **Q1. Performance degradation when two people are involved**
>
> It should be noted that two factors influence the identity of the actor.
>
> (1) The first factor is **the actor identification from the reference image**. In this paper, the customized action is decoupled from the actor in the reference image via a common action space (Figure 1, second and third rows).
>
> (2) Another factor is the **confusion between actors in the new context**, as previously mentioned with regard to Batman and Superman. It should be noted that even in the absence of consideration for customized actions, the stable diffusion model will nevertheless result in confusion regarding the identity information when generating multi subject[1,2]. The issue of how to eliminate confusion when generating multiple concepts represents a significant challenge, although it is not the primary focus of this paper.
>
> As a result, all models show performance degradation with two actors **due to the confusion of the multi-subject generation**.
>
> The issue of how to eliminate the confusion represents a significant challenge, although it is not the primary focus of this paper.
>
> [1]Easing Concept Bleeding in Diffusion via Entity Localization and Anchoring
>
> [2]FineControlNet: Fine-level Text Control for Image Generation with Spatially Aligned Text Control Injection
>
> ### **Q2. How the model can be customized for both character and action**
>
> The implementation of custom action generation can be integrated into existing custom production frameworks in an seamless manner. In the event that both the actor and the action are to be customized, it is only necessary to update the tokens of the action and the actor using different strategies. In particular, the actor tokens are updated directly by inversion, while the action tokens are generated by PCA coefficients, which are also updated by inversion.
>
> ### **Q3. Bad case analyses**
>
> One of the suboptimal results is the problem of confusion that arises when the generation process is conducted by multiple individuals. One potential solution is to introduce spatial constraints, such as bounding boxes[1] or manipulation attention maps, as discussed in reference [2]. It is also noteworthy that it is challenging to generate customized actions in certain specific contexts, such as when dealing with reptiles, such as snakes.
>
> [1] FineControlNet: Fine-level Text Control for Image Generation with Spatially Aligned Text Control Injection
>
> [2] Divide & Bind Your Attention for Improved Generative Semantic Nursing
>
> ### **Q4. Adding Figures**
>
> Thank you for your suggestions. We will readjust the layout in the final version to include the supplement of Figure 8 and Figure 9 in the paper, so that readers can have a better understanding

---

> > ### Comment · Reviewer_2ZgR · 2024-08-13
> >
> > Thank the reviewers for the rebuttal.
> > After a careful reading of the author's response as well as the comments from the other reviewers, my concerns have been clearly addressed. I decide to maintain the recommendation for acceptance of this paper.

---

> > > ### Author Response · Authors · 2024-08-13
> > >
> > > Thank you for your thoughtful consideration and for taking the time to engage with our responses. We are grateful for your recommendation for the acceptance of our paper.

---

### Official Review · Reviewer_Q179 · 2024-07-09

**Soundness:** 3
**Presentation:** 3
**Contribution:** 3
**Rating:** 6
**Confidence:** 4

**Summary:**

This paper aims to tackle the few-shot action image generation problem by customizing text-conditioned diffusion models. To decouple action from actor, the proposed method introduces a common textual action feature space to avoid the interference of the actor's visual semantics during action generation. The experimental results demonstrate the proposed method is effective in generating customized actions and preserving actor identities.

**Strengths:**

- This work focuses on customizing text-conditioned diffusion models for few-shot action image generation, which is an interesting and meaningful topic;
- The motivation of decoupling action and actor for action image generation is good, and the proposed textual action space is novel and effective in achieving the decoupling.
- The proposed method is easy to understand, and the writing is good.

**Weaknesses:**

The overall method is technically sound and the demonstrated results are great, but the reviewer still has some concerns about the method designs,

- The reviewer thinks it is reasonable to use three embeddings (i.e., k=3) to encode each action. However, it is hard to understand why the PCA is separately applied along the token embedding axis (e.g., when k=1, only all the first embeddings from each action are processed by PCA).
- The collected image action dataset is very small (12 actions only). Why not use some image/video action datasets? And, more importantly, how to make sure the learned/customized model can be generated to unseen action image generation.
- Regarding the action similarity loss, it is hard to ensure that the high-level action semantics will be learned, as there are no explicit constraints.
- minor issue: the high cost of action phrases filtering: the authors generate 1.2K action phrases using GPT-4 and a total of 2x1.2K images are generated using Stable Diffusion XL; The generation is expensive and the manual filtering process is labor-intensive.

/I will improve my rating if the author's response addresses all my concerns well.

**Questions:**

Please see the Weaknesses section.

**Limitations:**

The limitations are discussed and there is no potential negative societal impact.

---

> ### Author Rebuttal · Authors · 2024-08-07
>
> ### **For Reviewer** **Q179**
>
> We are sincerely grateful to the reviewers for dedicating their time and effort to review our work, and we appreciate the recognition of the novelty of our approach and the significance of our given the impressive result. We will try to address  reviewer's comments in detail below.
>
>
> ### **Q1. why the PCA is separately applied along the token embedding axis**
>
> We are grateful for your valuable suggestions. In response, we have included additional experimental results in the following table.  In our supplementary experiments, we focused on encoding action phrases at the phrase level, as opposed to the token level, using the CLIP model and leveraging the output of the final layer pool.
>
> | Methods    | S_Actor | S_Action |
> | ---------- | ------- | -------- |
> | w/ token   | 73.34   | 69.47    |
> | w/ phrases | 60.27   | 58.66    |
>
>
>
> The experimental results presented in Table 2 of the Rebuttal PDF indicate that conducting principal component analysis (PCA) on the embedding of individual tokens is more effective than performing PCA on the embedding of the entire action phrase. This is attributed to the greater flexibility afforded by analyzing the token dimension. For instance, when analyzing the phrase "raising hand," distinct coefficients can be assigned to the words "raising" and "hand" when PCA is applied to the token dimension. Conversely, applying varying coefficients to verbs and nouns across the entire phrase dimension is not feasible.
>
>
> ### **Q2.** **Enhancing Action Dataset for Unseen Action Image Generation**
>
> (1) Thank you for your suggestion, **w**e acknowledge that the current dataset is limited in scale. We will expand it further in subsequent work. However, we have made a concerted effort to ensure the diversity of the data by including various types of movements, such as localized movements, whole-body movements, and both single-person and two-person movements.
>
> Furthermore, the current dataset includes some actions that Stable Diffusion (SD) has not encountered before, such as those shown in **rows 1-4 of Figure 4 in the main paper**. Additionally, we have incorporated a more challenging action **in Figure 2 of the Rebuttal PDF.**
>
> (2) Collecting data from image or video action datasets is a good idea, but we find some datasets such as Stanford 40 Actions [1], ADHA [2], CMU MOCAP [3], tend to be designed for action recognition, meaning that most of them only contains **common actions** **(SD-know)** such as **running, jumping, playing guitar, chopping vegetables, etc.** Our goal is to fine-tune the stable diffusion model to generate images of **unseen actions**, so we focus on collecting images of actions that are **outside the distribution.**
>
> (3) For how to generate images of unseen actions, we have experimented in our paper. These experiments are illustrated **in rows 1-4 of Figure 4 in main paper and Figure 2 in the Rebuttal PDF.** These actions are unknown actions (as evidenced by the results shown in Figure 8 in the Appendix). The experimental results show that TwinAct is capable of generating these unseen actions effectively. Moreover, TwinAct can combine these actions with previously unknown actors, as demonstrated **in Figure 5 of the main paper (Actor OOD + Action OOD).**
>
> [1]http://vision.stanford.edu/Datasets/40actions.html
>
> [2]http://[www.mvig.org/research/adha/adha.html](https://www.mvig.org/research/adha/adha.html)
>
> [3]http://mocap.cs.cmu.edu/
>
> ### **Q3.The action similarity loss**
>
> (1)  The reconstruction loss is designed to **focus on the low-level details** of an image (see ADI[1], Section 1). However, it is prone to **overfitting** to the reference image (see Dreambooth[2]). To address this, we propose incorporating supplementary supervisory signals, specifically the semantic similarity of images.
>
> (2) CLIP, which has been pre-trained through image-text contrastive learning, has demonstrated **robust performance in numerous action recognition tasks** [3, 4]. Therefore, we utilize it as the action encoder.
>
> (3) The outcomes of the ablation experiments, as illustrated **in Figure 7(b) and row 4 of Table 2 in the main paper**, also indicate that the inclusion of non-low-level visual similarity loss can enhance the generation of customized actions.
>
> [1] Learning Disentangled Identifiers for Action-Customized Text-to-Image Generation
>
> [2] DreamBooth: Fine Tuning Text-to-Image Diffusion Models for Subject-Driven Generation
>
> [3] FROSTER: Frozen CLIP Is A Strong Teacher for Open-Vocabulary Action Recognition
>
> [4] ActionCLIP: A New Paradigm for Video Action Recognition
>
>
>
>
> ### **Q4. The high cost of action phrases filtering**
>
> The objective of our filtering process is to eliminate certain actions that are incapable of being accurately generated by SD. Consequently, our STEP for generating these images necessitates **25 steps** to ascertain the viability of action generation. In comparison, the typical number of STEPs for generating an image is 50, which expedites the generation process to a certain extent. The generation of the images for filtering required a total of **2.5 hours on an A100 GPU**.
>
> In the context of a manual filtering process, there are some possible **ways to reduce the cost**. The deployment of a multimodal large language model, such as GPT4, can serve as a filtering process. This approach entails the generation of images, which can then be subjected to filtering by GPT4. An additional method for reducing costs is to utilize an image description model. The filtering process entails the generation of corresponding text descriptions and the determination of whether these descriptions contain action phrases or a resemblance to the text prompt utilized for the generation of action images.

---

> > ### Comment · Reviewer_Q179 · 2024-08-10
> > **post-rebuttal-1**
> >
> > Thanks to the authors for their efforts during the rebuttal. After carefully reading the responses and comments from other reviewers, most of my concerns are well resolved, and I am willing to improve my rating.

---

> > > ### Author Response · Authors · 2024-08-10
> > > **Thanks to Reviewer Q179**
> > >
> > > Thank you sincerely for your thoughtful consideration of our rebuttal and the feedback from the other reviewers. We are pleased to hear that our responses have addressed your concerns and that you are willing to improve your rating.
> > > Please feel free to reach out if any further questions arise. We truly value your feedback and support throughout this process.

---

### Official Review · Reviewer_3NDQ · 2024-07-11

**Soundness:** 3
**Presentation:** 3
**Contribution:** 2
**Rating:** 5
**Confidence:** 2

**Summary:**

This paper introduced an text-to-action generation framework. In this framework, the authors first abstracted the actions (represented by natural language phrases) with PCA technique into a common action space. Then these high-level action features are fed into a text-to-image transformer network. In order to optimize the whole pipeline, the authors used a CLIP encoder to measure the cosine similarity between generated action images and reference action images.

**Strengths:**

1. The visualization results seem promising. Compared to other approaches in the submission, the introduced approach obtain better qualitative results.
2. The paper is well organized and easy to follow.

**Weaknesses:**

1. The authors claim that PCA is leveraged for feature compression when establishing common action space. Moreover, the authors borrow this technique from facial feature representations. It is known to us all that PCA is somehow an old-fashioned technique in the machine learning community. The authors are expected to provide more detailed explanations on why they choose this technique.
2. The network architecture is quite simple and lacks technical contributions. To the reviewer, the authors simply apply the off-the-shelf text-to-image framework in their pipeline. In the meantime, other components of the framework seems simple and weak in novelty.
3. The authors provide a similarity score as the evaluation metric. However, this score is computed by CLIP, which also serves as the action encoder during training. To the reviewer, the comparison seems not so fair. The consistency in IDs and actions should be more carefully evaluated to demonstrate the efficacy of the method.

**Questions:**

Please refer to weakness for more details.

**Limitations:**

The authors provide no limitations in the submission.

---

> ### Author Rebuttal · Authors · 2024-08-07
>
> ### **For Reviewer** **3NDQ**
>
> We appreciate your time and effort in reviewing our work, and we have carefully considered your comments. We will be sure to incorporate your suggestions to enhance the overall quality of the paper. We hope the following clarifications can address the reviewer's concerns:
>
> ### **Q1. More detailed explanations about PCA**
>
> We appreciate your insightful comments and concerns regarding our choice of Principal Component Analysis (PCA) to construct a common action space.  While PCA is a traditional technique, it is **still widely used** [1,2] today and has several advantages that make it appropriate for our study.
>
> (1) Its robustness to linearly correlated data, computational efficiency, and ease of interpretation fit well with our needs for feature compression and action space generalization.
>
> (2) **PCA is only used as part of the data preprocessing** in our method, and can be easily replaced by more advanced methods if needed.
>
> (3) To some extent, the superior performance of the common action space constructed based on the "old-fashioned" PCA algorithm in the experiment also verifies the effectiveness of our method.
>
> [1] Dual Prior Unfolding for Snapshot Compressive Imaging (cvpr2024)
>
> [2] Makeup Prior Models for 3D Facial Makeup Estimation and Applications (cvpr2024)
>
> ### **Q2. The network architecture**
>
> A straightforward and impactful approach is the goal we strive for in our paper. The innovative of this work can be summarized as follows:
>
> (1) **Insights into the relationship between action and actor confusion**: The domain of customized action generation remains relatively unexplored. In contrast to existing paradigms of contrastive learning, our approach introduces an action space as an effective method of biased induction, which decouples actions and actors.
>
> (2) **A novel custom token inversion method**: In contrast to typical customization techniques such as TI, dreambooth, and custom diffusion, our approach does not directly learn the parameters of custom tokens through inversion. Instead, we focus on learning the coefficients of action bases.
>
> (3) **Overcome the shortcomings in the understanding of reconstruction loss**: The original diffusion loss is not well-suited to the task of learning abstract action semantics from pixel reconstruction. Therefore, we introduce a loss function that measures image similarity, and the experimental results demonstrate its efficacy in improving customized action generation.
>
> In summary, our objective is to fine-tune a text-guided to image generative model. However, we propose new designs in multiple aspects, including **parameter initialization, fine-tuning manner, training paradigm**, and provide **detailed motivation, analysis, and ablation experiments** for each component. Extensive experiments also prove that TwinAct can achieve the best performance with its simple and intuitive design.
>
> ### **Q3. The evaluation metric**
>
> Thanks for your suggestion, we actually used two clips of different sizes. During training, we use **OpenAI CLIP ViT-L (246M)** as the action encoder, and **OpenCLIP ViT-bigG (1.39G)** to evaluate the similarity between generated and reference images. As you suggested, we supplemented the results used for the evaluation using other models (**Align [1]**) as shown in the Table 1 in Rebuttal PDF and below.
>
>
>
> | S_action | Text Inversion | DreamBooth | Reversion | Custom Diffusion |  P+   |  ADI  | Ours  |
> | :------: | :------------: | :--------: | :-------: | :--------------: | :---: | :---: | :---: |
> |   CLIP   |      9.12      |   12.23    |   18.73   |      26.83       | 33.95 | 45.32 | **69.47** |
> |  Align [1]   |     10.05      |   11.67    |   17.37   |      28.33       | 30.45 | 44.93 | **70.82** |
>
>
>
> The experimental results show that TwinAct **still achieves the best results** among the Align-based evaluation results. In addition, we also constructed **user study (Table 1 in main paper)** and **4-dimensional error analysis (Figure 4 in main paper)**. These experimental results also prove the superiority of TwinAct.
>
> [1] Scaling Up Visual and Vision-Language Representation Learning With Noisy Text Supervision

---

> ### Author Response · Authors · 2024-08-12
>
> Dear Reviewer 3NDQ:
>
> Thank you for your feedback. We have carefully addressed your comments in our rebuttal and kindly ask you to review them at your earliest convenience.
>
> If you have further questions, please do not hesitate to contact us—we are committed to providing thorough responses.
>
> Thank you once again for your time and effort in reviewing our work. We look forward to your response.
>
> Best regards

---

> ### Comment · Reviewer_3NDQ · 2024-08-13
>
> Thanks for the authors' responses. I found myself learning towards the responses, despite my initial hesitations.

---

> > ### Author Response · Authors · 2024-08-13
> >
> > We greatly appreciate your openness and are glad that our clarifications were helpful. We are grateful for your support and feedback.

---

### Official Review · Reviewer_BXKD · 2024-07-13

**Soundness:** 3
**Presentation:** 3
**Contribution:** 3
**Rating:** 6
**Confidence:** 4

**Summary:**

Problem: Preserve the consistency of the reference action when generating images with new actors by decoupling the action and actor properly.

Key Idea:
The authors propose a method to disentangle actions from actors in text-guided diffusion models and generate new images that exactly replicate the action pose in the given few-shot inputs. The main contributions are:

-	Generating action phrases using GPT-4 and filtering 832 action phrases know to the model. These are embedded into a common action space with a set of action bases, thus containing no distractions from non-action related information.

-	The action bases are then combined using a multi-layer perceptron to create new customized actions. Instead of backpropagation to determine optimal coefficients, the authors use CLIP as an action decoder to extract semantic features.

-	An action similarity loss is introduced between the encoded features of the reference and generated action to imitate the high-level action more accurately.

-	A novel benchmark consisting of 12 actions to compare with existing methods. The authors also provide extensive analysis and ablations.

**Strengths:**

-	The proposed method replicates the sample action more precisely with new actors whereas existing methods make mistakes in the details. This shows that the method has successfully isolated the action information to apply it for new image generation.

-	The examples in the provided figures give a clear idea where other methods are failing while the proposed one successfully replicates the input action with a new actor.

-	The authors report higher scores than existing methods based on both objective and user study.

-	The ablation studies show the impact of each contribution and varying the number of principal components (p).

**Weaknesses:**

-	The identity of actors is preserved is a strong claim which is not reflected in qualitative evaluation. For example, in Figure 4 row 4: the faces of Batman and Superman look similar. In rows 5 and 6, the subject does not exactly look like Leonardo. In the last row, we see Spiderman in place of Superman.

-	In some cases, the generated image contains a mirrored action of the sample image. For example, in Figure 5 row 2, the generated action is performed with the opposite hand than the sample action in the 3rd and 4th images.

-	The output generation space is limited, so some complicated customized actions might not be replicable by combining the bases.

-	It is not clearly shown how changing the value of coefficients for the same action bases impact the generated image.

-	Currently the model has pre-trained knowledge of only 25 actors and cannot be easily adapted to new ones without retraining.

**Questions:**

- How is the CLIP score exactly being calculated for SAction and SActor? Is it simply an image-to-text matching score?

- For the human evaluation, what is the degree of knowledge of the 100 users (average or expert)?

- How do you plan to scale this method for a higher number of action variations? Just generating more actions using GPT-4 and adding their embedding to the action space might not be enough.

- Is it possible to show examples of what happens if we vary the value of coefficients in the action bases and how they impact the generated image? For example, a crouching position might be a combination of action bases related to sitting and standing. So, if the coefficient of sitting is 0, the generated image should only contain standing. Again, the image might only contain sitting in the opposite case. The intermediate values of these will show various angles of crouching.

- Providing a TSNE visualization of the action space might also help in clarifying.

- For multiple actors, the authors show examples of images containing only two subjects. Can this be increased to more subjects?

- Also, for multiple actors, how is the role in the image defined? For example, in Figure 4 last row, how will we generate Leonardo carrying Obama or Barbie carrying gorilla?

-- POST rebuttal: the authors have addressed all the raised concerns, increasing my rating.

**Limitations:**

The authors addressed the concerns of limited output possibilities of their common action space and physical defects of TGDMs. They have not discussed any other limitations.

---

> ### Author Rebuttal · Authors · 2024-08-07
>
> ### **For Reviewer** **BXKD**
>
> Thank you for recognizing our paper and recommending it for acceptance. Now, we will address the key arguments raised in the reviews.
> ### **Q1.The identity of actors**
>
> It should be noted that two factors influence the identity of the actor.
>
> (1) The first factor is **the actor identification from the reference image**. In this paper, the customized action is decoupled from the actor in the reference image via a common action space (Figure 1, second and third rows).
>
> (2) Another factor is the **confusion between actors in the new context**, as previously mentioned with regard to Batman and Superman. Even in the absence of consideration for customized actions, the stable diffusion model will nevertheless result in confusion regarding the identity information when generating multi-subject [1,2]. The issue of how to eliminate confusion when generating multiple concepts represents a significant challenge, although it is not the primary focus of this paper.
>
> In order to maintain the rigor of the paper, we will modify the description of how the actor's identity is maintained in the paper.
>
> [1]Easing Concept Bleeding in Diffusion via Entity Localization and Anchoring
>
> [2]FineControlNet: Fine-level Text Control for Image Generation with Spatially Aligned Text Control Injection
> ### **Q2.The generated image contains a mirrored action of the sample image**
>
> The reason for generating the action containing the mirror is twofold:
>
> (1) One is that the actions in different directions are **contained in the reference image**.
>
>  (2) The other is because we applied **image augmentation** (including rotation) to the training data.
>
> We consider that the mirrored actions are, at the very least, not semantically incorrect. However, should one desire to generate actions that are entirely customized, it would be advisable to consider removing the aforementioned two factors.
> ### **Q3.Some complicated customized actions might not be replicable**
>
> (1)In this paper, we have endeavored to conduct a comprehensive evaluation of TwinAct's capabilities, encompassing a range of scenarios. These scenarios **include detailed hand movements, full-body movements involving multiple body parts, and movements requiring the coordination of multiple individuals.**
>
> (2)We added **a complex action as shown in Figure 2** in Rebuttal PDF and the results demonstrate the superiority of the TwinAct method.
>
> (3)In the case of particularly intricate actions, the complexity of the customization process can be mitigated by **broadening the scope of the common action space** and **augmenting the number of reference images**.
> ### **Q4.Changing the value of coefficients for the same action bases**
> We have made adjustments to the coefficients as suggested and have generated intriguing findings as shown **in Figure 1 in Rebuttal PDF**.
> ### **Q5.** **Easily adapted to new ones**
>
> Although the focus of this paper is on the generation of customized actions, rather than customized actors, it also illustrates the capacity of TwinAct to synthesize SD-known actors but also presents experimental findings on a personalized actor dataset (**Figure 5 in the paper**), demonstrating that TwinAct can be easily integrated into existing methods for customizing actors.
>
> Furthermore, the customization of an action by TwinAct is a relatively expeditious process, requiring approximately only **15** minutes.
> ### **Q6.  Calculated for SAction and SActor**
> In the case of S_action, our approach aligns with that of previous customization work[1,2]. The custom diffusion process employs the image from the clip and the image similarity metric for evaluation. In the case of S_actor, the pre-trained face encoder is employed to ascertain the degree of similarity.  Furthermore, to provide a more comprehensive evaluation of the generated results, we have conducted a user study.
> [1] Multi-Concept Customization of Text-to-Image Diffusion
> [2] DreamBooth: Fine Tuning Text-to-Image Diffusion Models for Subject-Driven Generation
> ### **Q7. The degree of knowledge of the 100 users**
> We employ 100 users from the outsourcing platform to conduct user research. The knowledge level of users includes junior college and undergraduate, and the ratio of junior college to undergraduate college is close to 2:1. We also provided detailed documentation and examples to ensure that all participants clearly understood the purpose and requirements of the task.
> ### **Q8. Scale this method for a higher number of action variations**
>
> If the action space is to be further extended:
>
> (1) one feasible way is to add new action phrases through expert knowledge. For example, it is possible to expand the action space by dividing the body parts at a more detailed level of granularity, such as the eyes, the mouth, the thumbs, and so forth, in order to realize a greater number of combinations of actions.
>
> (2) another possible way is to add customized actions token embedding generated by our methods to the action space as well, which we leave as an exploration for future work.
> ### **Q9. TSNE visualization of the action space**
> We have provided visualization results of T-SNE for the action space. The objective of our work is to generate customized actions through action-based combination, with the optimal action base being orthogonal. As shown **in Figure 4 in Rebuttal PDF**, the action base does not exhibit discernible clusters, and the distribution is relatively uniform, which aligns with the nature of our action base.
> ### **Q10. Increased to more subjects**
> Thanks for your suggestion. We additionally present the generated results of customized actions involving 3 people, as shown **in Figure 2 in Rebuttal PDF**. These results effectively demonstrate TwinAct's capability to manage multiple actors.
> ### **Q11. how is the role in the image defined?**
> If the role in the image needs to be specified, it can be specified by prompts, as shown **in the Figure 3 in Rebuttal PDF.**

---

> > ### Comment · Reviewer_BXKD · 2024-08-13
> > **After author response**
> >
> > Thank you for providing a very detailed response, it addressed most of the concerns. I do have a couple of more questions/ just for clarification,
> > - The shown t-sne in the rebuttal pdf does not look good, it is not clear, with such good generated samples, why clustering would be so bad? Are the shown samples cherry picked? or I am missing something here?
> > - I think, it will not be good to claim identify preservation, considering it is not the main focus and also there is no strong evidence.
> > - It is not clear why rotation augmentation will lead to flipped actions?
> > - I think, claiming the proposed approach can do complex actions will be a stretch, although the shown samples look reasonable, it is not clear how the spatial ordering, role, etc, can be assigned, even using language, the shown samples seems like random assignment.

---

> ### Author Response · Authors · 2024-08-13
> **Reviewer BXKD**
>
> Thank you for your thoughtful comments. We sincerely appreciate your feedback. To address your queries, we have provided further explanations below:
>
>
> ## **Q1 The clustering in t-sne**
>
>
>
> We must point out that the purpose of an action base is to combine new actions. This suggests that it is desirable for these action bases to **be orthogonal to each other**.  In other words, **each action base should be unique** in the action space so that more new actions can be combined.
>
> Therefore, when constructing the action space, we **filter repetitive or similar action phrases** to ensure the independence/diversity of the action base.
>
>
>
> For orthogonal bases, t-SNE projects them to different locations (e.g., uniform distributions) so that these basis vectors are separated from each other in low-dimensional space. Thus, these basis vectors can be more flexibly combined to generate new vectors(i.e. new actions).
>
>
>
> On the other hand, if the action bases form mixed clusters in the action space, indicating that they are highly correlated with each other, such basis vectors will show poorer combining ability. Thus our clustering is not bad.
>
>
>
>
>
>
> ## **Q2 The stability of the generated samples**
>
>
>
> In addition to the qualitative visualization results, we also present **quantitative experimental results** in the paper, and several experimental results demonstrate the superiority of TwinAct. In the **user study**, users also expressed a clear preference for images generated by TwinAct.
>
>
>
> Due to the limitation of the discussion, we cannot add more visualization results, but we will **add the results generated by multiple random seeds** in the final paper to further prove the stability of the generated results. Finally, our dataset and code will be open source.
>
>
>
>
>
> ## **Q3 Identify preservation**
>
>
>
> Thank you for your suggestion. We will revise the presentation about identify preservation in the final version. **Decoupling the action from the actors** may be more accurate.
>
>
>
>
>
> ## **Q4 About the rotation augmentation**
>
>
>
> We apologize for the loose wording, but the "rotation" we refer to in data augmentation includes both small rotations (**RandomRotation(degrees=30)**) and large rotations (**RandomHorizontalFlip**).
>
> Specifically, for image augmentation, we use the following techniques: **RandomHorizontalFlip, RandomRotation, RandomAffine, and ColorJitter**, which are randomly selected with a probability of 50%.
>
>
>
>
>
> ## **Q5 About role assignment**
>
>
>
> For the carry example, since it has an **explicit active-passive relationship**, we can specify it through language. Specifically, we have a mixture of "A v_carry B" and "B is v_carry A" in our dataset, so the model can learn that **"v_carry" means that the front character is always standing**, and **"is v_carry" means that the front character is always lying down**. So we can use "Leonardo v_carry Obama" or "Obama v_carry Leonardo" to specify who is standing in the back and who is lying in the front
>
>
>
> However, we recognize that it is difficult to linguistically constrain the spatial constraints for other examples that do not have an obvious active-passive relationship, such as the actions in rows 3 and 4 in Figure 1 of the main paper, and we tried "in the left" and "in the right," but SD does not work for this type of spatial constraint. We tried "in the left" and "in the right", but SD has a limited understanding of this type of spatial relationship. The generated roles are random.
>
>
>
> A possible solution is to **add layout information** such as a bounding box for control, and our method can seamlessly access similar methods such as controlnet. **We will add the generation results of introducing ControlNet[1] in the final version.**
>
> [1] Adding Conditional Control to Text-to-Image Diffusion Models
>
>
>
> Thank you for your valuable feedback on our paper, we will carefully consider your suggestions and revise the final manuscript.  In addition, we would sincerely appreciate it if you could possibly raise your score.
>
> Finally, please feel free to contact us if you have any further questions. We look forward to your response.

---

> > ### Comment · Reviewer_BXKD · 2024-08-13
> >
> > Thank you for clarifying the doubts. All my concerns have been addressed. I will increase my rating.

---

> > > ### Author Response · Authors · 2024-08-13
> > >
> > > Thank you so much for your understanding and support! We are glad to hear that all your questions have been addressed.

---

### Author Rebuttal · Authors · 2024-08-07

### For ALL Reviews

We sincerely thank all the reviewers for their thoughtful feedback. We are glad to see that most of the reviews have recognized our work:

**[ALL Reviews]** **Robust and Superior Results:** We are grateful for the reviewers’ positive feedback on our impressive experimental results, particularly our method's ability to consistently outperform existing approaches in both objective metrics and user studies.

**[BXKD, Q179, 2ZgR]** **Innovative Approach:** We are delighted that most reviewers acknowledge our motivation is good and interesting, and the novelty and effectiveness of our method in action image generation.

**[ALL Reviews]** **Detailed and Accessible Presentation:** We appreciate the reviewers highlighting the comprehensiveness of our explanations and the ease of understanding our paper.

We will now address the key points raised in the reviews and provide detailed responses to each reviewer. We highly recommend that reviewers take the time to **check PDF for additional visualization of results**.

Once again, we extend our thanks for the reviewers’ time and valuable insights, and we look forward to any additional feedback or questions regarding our work.

---

> ### Comment · Area_Chair_qyvf · 2024-08-08
> **Author-Reviewer Discussion Begins!**
>
> Dear authors,
>
> Thanks for your hard work and detailed responses.
>
> Dear reviewers,
>
> the authors have responded to your questions and comments, please read them and further provide feedback: Whether your concerns addressed? After reading all the reviews and responses, are there more questions about the clarity, contributions, results, etc?
>
> Thanks!
>
> Best, Your AC

---

> > ### Comment · Area_Chair_qyvf · 2024-08-12
> > **Reminder: author-reviewer discussion will end soon!**
> >
> > Dear reviewers,
> >
> > The authors have responded to your questions and comments. Please propose your post-rebuttal comments.
> >
> > Thank Reviewer Q179 for your positive discussions!
> >
> > Best, Your AC

---

### Decision · Program_Chairs · 2024-09-25

**Decision:**

Accept (poster)

**Comment:**

The paper proposes a method to decouple actions and actors for few-shot action image generation in text-guided diffusion models. It has three key steps: building the common action space based on representative action phrases, imitating the customized action within it, and generating customized action images with action similarity loss. Three of the reviewers' concerns are addressed and one reviewer's most concerns are addressed. After reading the paper, reviews, and responses, the AC agrees that this paper has proposed a sound method, and the method has shown interesting improvements.